

# Evapotranspiration Dynamics in Monsoon-dominated Region in the Korean Peninsula

Eunji Kim[1], Boosik Kang[2]

[1]K-water (Korea Water Resources Corporation), Daejeon, 34350, the Republic of Korea

[2]Dept. of Civil and Environmental Engineering, Dankook University, Yongin,16890, the Republic of Korea

*Correspondence to*: Boosik Kang (bskang@dankook.ac.kr)

**Abstract.** This study investigates the dynamics of evapotranspiration in a monsoon-dominated region of the Korean Peninsula, with a particular focus on the challenges associated with measurement, identification, and prediction of the potential and actual evapotranspiration. The research delves into various models and theories, notably the Complementary Relationship of

Evapotranspiration (CRE) hypothesis widely used for estimating AET indirectly when direct measurements are difficult to obtain. The study area encompasses the Yongdam dam basin, utilizing data from flux towers, evaporimeters, and meteorological stations to estimate both actual evapotranspiration (AET) and potential evapotranspiration (PET). The Penman-Monteith equation is employed for PET estimation, with reference evapotranspiration calculations conducted using the FAO Penman-Monteith equation. This research confirms the existence of complementary relationship behaviour in regions where

strong correlations between soil moisture and air humidity are observed, such as deserts and tropical areas. In these regions, the influence of annual climate fluctuations and seasonal winds is comparatively minor. Nevertheless, it is important to note that the correlation between soil moisture and air humidity diminishes in areas affected by external factors, such as the dominant influence of the monsoon climate zone. In such instances, potential evaporation and actual evaporation often deviate from the expected complementary relationship, adopting a more erratic pattern of distribution.

## 1 Introduction

The global or local water cycles are largely affected by climate components, e.g., precipitation, atmospheric temperature and evapotranspiration, etc. which are most directly attributable to climate change. While the vertical and lateral water budget through land surface and atmospheric interactive system is critically important in not only understanding hydrologic cycle itself but also validating the atmospheric model performance, the spatial and temporal distribution of the evapotranspiration is

hard to be measured, identified and predicted. The net transport of moisture from ocean to land, equivalent to 3.2 Petawatt of energy annually, significantly impacts monsoon dynamics and global circulation patterns (Trenberth et al., 2009). As such, evapotranspiration accounts for a large part of the total water budget in the hydrologic cycle, which is an important factor to be considered in the management and planning of water resources. Evaporative dynamics exhibit complex behaviours



depending on the degree of saturation of the land surface and ambient air as well as incoming solar radiation, and especially

in areas with strong seasonal effects, such as the monsoon climate zone, variability tends to be higher(Bohn and Vivoni, 2016). Monsoon refers to the phenomenon where the wind direction changes due to the temperature difference between the continent and the ocean in summer and winter. Local meteorological phenomena manifest diversely due to the contrast in thermal properties between the sea and land, resulting in land breezes, topographical influences causing mountain and valley winds, and the development of urban heat islands due to physical differences in surface features between rural and urban areas, unlike

synoptic-scale atmospheric phenomena. The Korean Peninsula, with its complex geographical conditions, exhibits varied climate characteristics across different regions. This implies the need for detailed climate analysis associated with the monsoon climate throughout the entire peninsula.

Quantifying the actual evapotranspiration is most accurately achieved through flux measurement, yet the establishment of a dense observational network poses considerable challenges. Since it is technically limited to measure the actual

evapotranspiration (AET) over the large-scale watershed, basin or continent for operational purposes, methods for estimating it through indirect calculation were tried by Oldekop (1911), Thornthwaite (1948), Penman (1948), Bouchet (1963), Budyko (1974), and Brutsaert and Stricker (1979), etc. Oldekop (1911) was the first to establish the aridity index in his formula relating streamflow to precipitation and evapotranspiration. In the middle 1940's, breakthroughs were made in the estimation of evaporative demand and potential evapotranspiration (PET) (Thornthwaite, 1948; Penman, 1948). They parameterized

evaporative limits simply on a few meteorological components, e.g., air temperature, net available radiative energy, drying power, etc., through a one-way approach. The first consideration of the interactive relationship between land surface and ambient atmosphere linking both the supply of and demand for the evaporative moisture was suggested by Bouchet (1963) as hypothesizing the complementary relationship of evapotranspiration (CRE) between variations in regional-scale AET and those in point-scale of evaporative demand.

The Bouchet's hypothetical theory has been discussed with respect to the regional evaporation, highlighting the interaction between actual and potential evaporation rates under various climatic conditions. . Numerous studies, based on the Advection-Aridity (AA) model proposed by Brutsaert and Stricker (1979) and the Complementary Relationship of Areal Evapotranspiration (CRAE) model developed by Morton (1983) have been carried out for identifying implications for the hydrological cycle, climate change and attributes from the factors such as reduced solar radiation, wind speed, and vapor

pressure deficit (Brutsaert and Parlange, 1998; Golubev et al., 2001; Ozdogan and Salvucci, 2004; Yang et al., 2006; Pettijohn and Salvucci, 2009). Despite the numerous theories on evapotranspiration, empirical verification in actual basins has been quite limited due to insufficient spatial observation density and data quality.

Hobbins et al. (2001a) examined the discrepancies between potential evapotranspiration (PET) and actual evapotranspiration (AET) measurements across 120 watersheds in the United States. By analyzing long-term data, the authors highlight the

influences of climate change and variability on water resource availability, emphasizing the importance of accurate PET and AET estimations for effective water management. Their results indicated that the CRAE model tends to overestimate, whereas the AA model tends to underestimate AET, necessitating parameter adjustments for the AA model in arid regions. In a





subsequent study, Hobbins et al. (2001b) re-evaluated the Priestley-Taylor parameter (α) from Priestley and Taylor (1972) and addressed the issue of AET underestimation by the AA model. Ramirez et al. (2005) aimed to provide empirical support for

Bouchet's hypothesis on the complementary relationship in regional evaporation by analyzing observational data. However, their study was constrained by the use of historical observational data, which may contain measurement inaccuracies and spatial variability and the applicability of Bouchet's hypothesis to diverse climatic conditions beyond the studied regions remains uncertain without further validation.

As previously mentioned, observational data sets with concurrent time and location for potential evapotranspiration (PET) and

AET are rarely available. The CRE hypothesis was validated by using annual-scale observational data, defining PET as pan evaporation ($E_{pan}$) and AET as the difference between annual precipitation and runoff. Xu and Singh (2005) evaluated different methods for estimating potential evapotranspiration (PET), e.g. CRAE, AA, and GG models, comparing their accuracy and applicability across various climatic regions. Ma et al. (2015) investigated the trends and driving factors of evapotranspiration changes in the Loess Plateau, China, over several decades. It identified land use changes and climate variability as key

influences on the observed evapotranspiration patterns, offering insights for regional water resource planning.

Kahler and Brutsaert (2006) and Zuo et al. (2016) recently suggested that $E_{pan}$ exhibits an asymmetrical CRE with. They examined the asymmetrical CRE and variability of reference evapotranspiration in China from 1961 to 2013, considering the impacts of climate change. The authors identify significant regional differences in evapotranspiration trends, influenced by factors such as temperature, solar radiation, and wind speed, highlighting the complexity of climatic effects on

evapotranspiration.

Because the observation of AET and WET can be limited available, in several studies, the AET is usually estimated indirectly from water balance analysis among precipitation, rainfall and AET. Lameur and Zhang (1990) calculated AET by applying AA and CRAE models in dry areas and compared them with AET based on the water balance method. Liu et al. (2006) applied AA, CREA, and GG models to the Yellow River basin and evaluated the applicability of the models through comparison with

the basin-wide water balance-based ETA. Granger and Gray (1990) calculate the evapotranspiration by adding precipitation and soil moisture change and compare it with the results of the CRAE model.

Nevertheless, considering that the complementary theory is a hypothesis rather than a manifestation based on physical laws, validations has cantered around the intricate associations among potential, actual, and wet-environment evapotranspirations. Verification endeavours have been continued, encompassing diverse locales and scales. Despite the ongoing nature of this

validation process, there remains a dearth of scholarly attention directed towards discerning the constraints inherent in the complementary theory. This current study aims to scrutinize the dynamic interplays involving solar radiation, atmospheric vapor pressure, flux, potential and wet-environment evapotranspiration within the monsoonal region of the Korean Peninsula. The objective is to delineate both the limitations and the applicability of the hypothesis of the complementary relationship under the monsoon climate environment.





## 2 Study area and data sets

### 2.1 Study area


The study area is the Yongdam dam basin, which is located upstream of the Geum river basin (Fig. 1) in South Korea. The total area of basin is 930 km2, and the forest account for 79.9% (743.4 km2) of the total area. The highly reliable measuring unit for actual evapotranspiration is eddy covariance flux tower. Flux is the total amount of air moves through a unit area in a

unit time, and since it varies depending on the intersection area and time, it is the mean product of the vertical wind speed and the value of the property. It depends on several properties i.e., crossing area, size of area of interest being crossed and time takes to cross the area. Eddy covariance requires measurements of 3-dimensional wind speed and gas concentration. For accurate determination of the air flux the high speed and precision instruments are most critical for rapid measurement of all the small changes in the air samples.

There are two flux towers (Yongdam and Mt. Deogyu), one evaporimeter (large evaporation pan) on the surface of dam, and three Meteorological stations (Geumsan, Jangsu, and Jeonju) located in or around the Yongdam dam basin managed by the Korea Meteorological Administration (KMA) (Figure 1). The location of the flux towers and the weather stations is presented in Table 1-2. The Yongdam dam site has the data set for only two years of 2017 and 2019 whereas the Mt. Deogyu site has relatively long years of operation period from 2011 to 2019. There are missing flux data in 2018. The Mt. Deogyu flux tower

is located at the site having NDVI(Normalized Difference Vegetation Index) higher (0.33) than basin average (0.24). The elevation of flux Tower is EL. 688.568 m and most of the land use in the watershed is forest, and the clinical distribution includes invasive mixed forests and larch forests. The flux tower was installed with a 19 m tower in consideration of the impact of the water body, the height of the vegetation canopy, and accessibility. The 3D sonic anemometer (CSAT3, Campbell Scientific, Logan, USA) measuring wind speed in the three directions and the open path gas $H_2O/CO_2$ analyser (LI-7500, Li-

Cor Biosciences, Nebraska, USA) measuring concentrations of water vapor and CO2 at a 20 Hz-frequency were installed at the top of the tower.



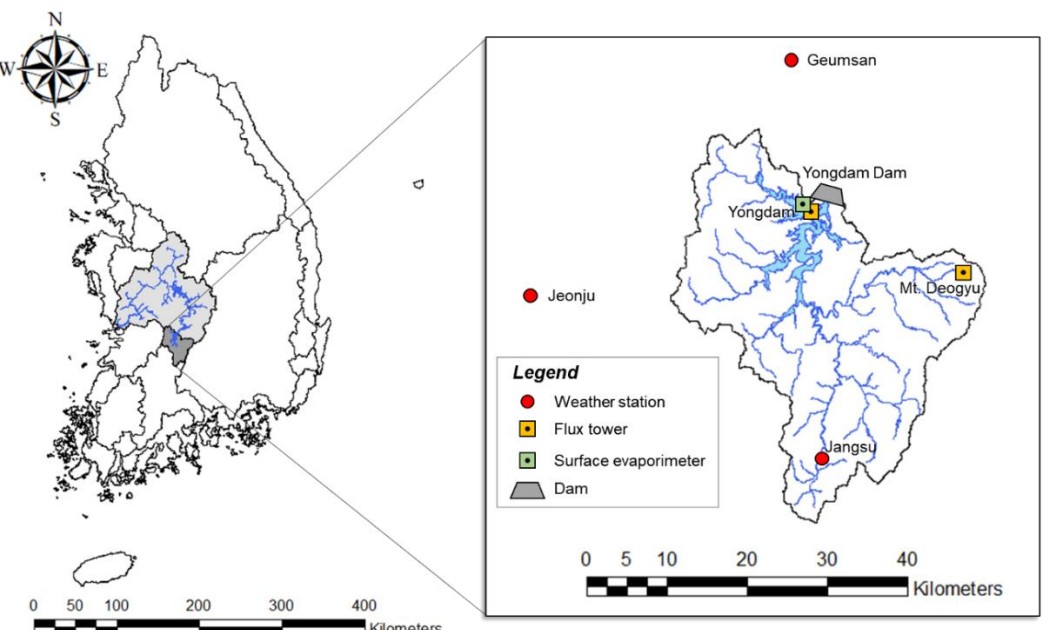

**Figure 1** Yongdam dam basin map and location of flux towers and meteorological stations

**Table 1.** *Location of the flux towers and the evaporimeter on the surface of Yongdam dam*

| Name | Latitude | Longitude | Altitude | Operations Start Date |
|---|---|---|---|---|
| Mt. Deogyu | 35° 51' 53" | 127° 43' 02" | 688.6 m | 2011. 4. 11. |
| Yongdam | 35° 56' 10" | 127° 30' 20" | 263.6 m | 2017. 4. 20 |
| Surface Evaporimeter | 35° 56' 31" | 127° 31' 14" | - | 2014. 3. 1 |

**Table 2.** *Location and observation information of the meteorological stations around Yongdam dam basin*

| Station | Latitude | Longitude | Altitude (EL.m) | Ground height of equipment (m) | | | Operations Start Date |
|---|---|---|---|---|---|---|---|
| | | | | Manometer | Thermometer | Rain gauge | |
| Geumsan | 36° 6' 20" | 127° 28' 54" | 173.7 | 174.0 | 1.5 | 0.5 | 1972. 1. 9 |
| Jangsu | 35° 39' 25" | 127° 31' 13" | 406.9 | 408.0 | 1.6 | 0.5 | 1988. 1. 1 |
| Jeonju | 35° 50' 27" | 127° 7' 7" | 61.4 | 62.9 | 1.6 | 1.4 | 1918. 6. 23 |




## 2.2 Hydrologic data description

The meteorological components measured in those weather stations include daily meteorological data; precipitation (mm/day), temperature (℃), wind speed (m/s), relative humidity (%), actual vapor pressure (kPa), and atmospheric pressure (kPa). Solar radiation (MJ/m2/day) and pan evaporation ($E_{pan}$) (mm/day) data are provided only at the Jeonju meteorological station. In addition, the daily dam inflow data was obtained from the Water Resources Management Information System (WAMIS) of the Ministry of Environment, South Korea.

The annual precipitation in the Yongdam dam basin during 2012 to 2019 is 1358.0mm. The annual actual evapotranspiration measured from the flux tower at Mt. Deogyu is 333.2mm and annual pan evaporation is 1047.4mm during the same period. The daily plots for the flux, PET estimated by the Penman-Montheith equation corresponding to pan evaporation and WET estimated by the Priesley-Taylor equation are shown in Figure 2-4. Both ETs show typical seasonal variation appearing in monsoon climate region. The upper limit of flux shows changes in the seasonal cycle, and overall, it shows a high degree of

scattering below the upper limit. Unlike flux, PET shows seasonal oscillations of upper and lower limits, and shows higher scattering in summer than in winter when radiant energy is strong and weather changes are severe.

It is very important to predict the AET in order to understand the natural water budget and to manage water resources. However, as shown in Figure 2, the flux has a much higher scatteredness than the PET or WET, and the flux sensor using eddy covariance has difficulty in measuring and predicting because of the lack of observation density. Until now, predicting the AET using the

complementary relationship theory proposed by Bouchet(1963) and developed by Budyko(1974) is one of the most referred methodology for predicting the actual evapotranspiration(Xu and Singh, 2005; Fu, et al., 2023)

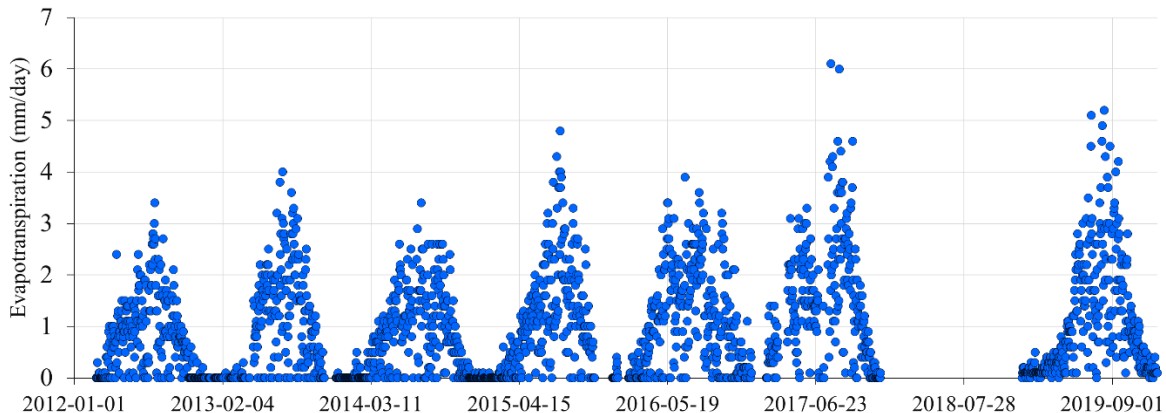

**Figure 2** Time Series plot for the Evapotranspiration Flux during 2012 to 2019 at Mt. Deogyu



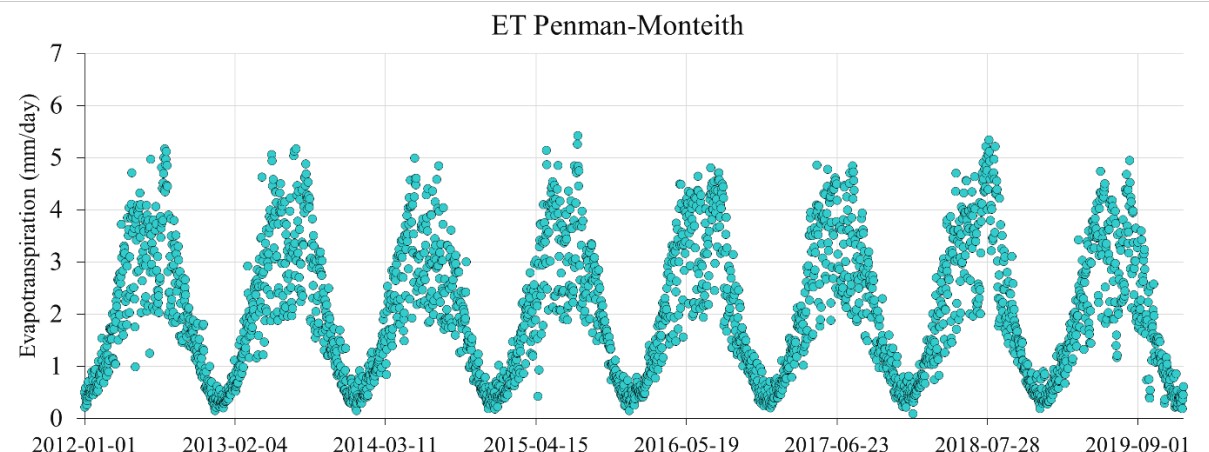


**Figure 3** Time Series plot for the PET during 2012 to 2019

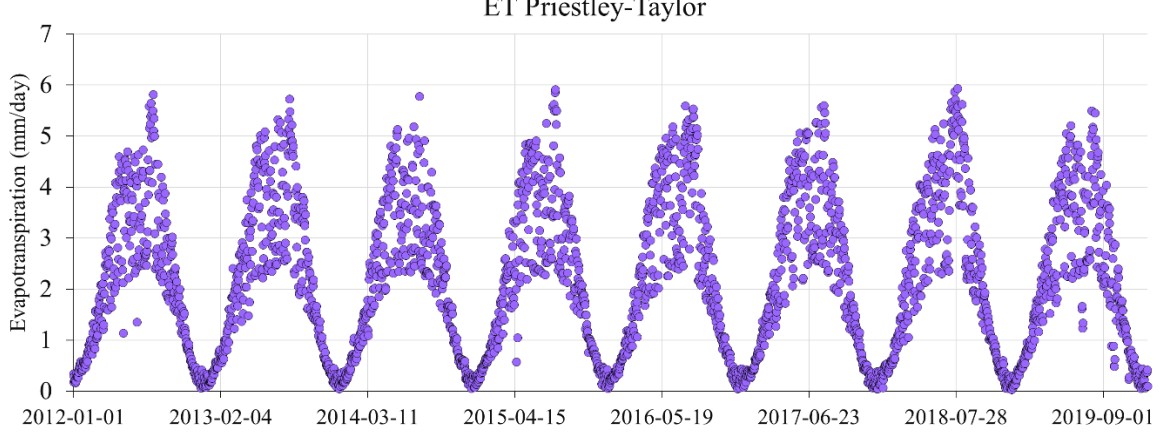

**Figure 4** Time Series plot for the WET during 2012 to 2019

## 3 Complementary relationship of evapotranspiration (CRE)

### 3.1 Bouchet's CRE hypothesis

Bouchet's Complementary Relationship in Evaporation (CRE) hypothesis, fundamentally redefined the understanding of the interplay between actual evaporation (E) and potential evaporation ($E_p$). Bouchet posited that in a given region, the sum of actual evaporation and potential evaporation remains nearly constant (Bouchet, 1963). This relationship suggests that when actual evaporation decreases due to limited water availability, potential evaporation increases, and vice versa. This

counterintuitive interaction hinges on the principle that changes in surface moisture conditions and atmospheric demand create a balance, where a deficit in one component leads to a compensatory response in the other.





Bouchet illustrated this hypothesis with the concept that potential evaporation represents the evaporation rate under unlimited water supply, while actual evaporation reflects the reality constrained by available moisture. In periods of ample water supply,
actual evaporation approaches potential evaporation, minimizing the difference between them. Conversely, during dry periods, actual evaporation drops, but potential evaporation rises due to increased energy and vapor pressure deficit, maintaining a near-constant total.

Brutsaert and Stricker (1979) built upon Bouchet's hypothesis by developing a quantitative framework, the Complementary
Relationship Areal Evapotranspiration (CRAE) model. They introduced mathematical formulations (Equation [1]) to describe the complementary relationship more precisely, linking actual evaporation, potential evaporation, and pan evaporation ($E_{pan}$).

$$AET + PET = 2WET \qquad [1]$$

The Brutsaert and Stricker's model provided empirical methods to test and validate the CRE hypothesis, bridging theoretical
concepts with practical applications. It emphasized that the complementary relationship is governed by the interplay between surface moisture availability and atmospheric demand, reinforcing Bouchet's idea that these factors are inversely related. Their contributions were expected to have the applicability of the CRE hypothesis to various climatic conditions and regions, allowing for improved estimation of regional evapotranspiration. The Bouchet's original hypothesis and Brutsaert and Stricker's advancements laid a critical foundation for contemporary studies in hydrology and climatology, offering a
comprehensive perspective on the dynamics of evaporation.

Granger (1989), however, proposed a method for estimating evapotranspiration by incorporating both surface resistance and aerodynamic factors, implicitly addressing asymmetries in the evaporation process. It highlights the role of these asymmetric factors in accurately capturing the dynamics of evapotranspiration under varying environmental conditions. Subsequent studies on asymmetric CRE have led to the formulation of Equation [2] and tried to improve the reliability of regional
evapotranspiration estimates, especially in diverse climatic environments (Brutsaert and Parlange, 1998; Pettijohn and Salvucci, 2006; Szilagyi, 2007).

$$b(WET - AET) = PET - WET \qquad [2]$$

where b is the coefficient for asymmetric CRE.

Figure 5 is a schematic diagram of the CRE equations of several researchers mentioned above. The CRE equation is expressed
as the dimensionless equation. In this study, the AET and PET are divided by the WETs and defined as the AET* and PET*, respectively, and these variables are represented on the vertical axis. The moisture index is defined as the AET divided by the PET and expressed on the horizontal axis. The Figure 5(a) is the symmetrical CRE relationship and the PET* and AET* decreases or increases at a same rate above or under the WET. However, the Figure 5(b) is expressing Eq. [2] and shows the variation of PET* and AET* according to the parameter b.





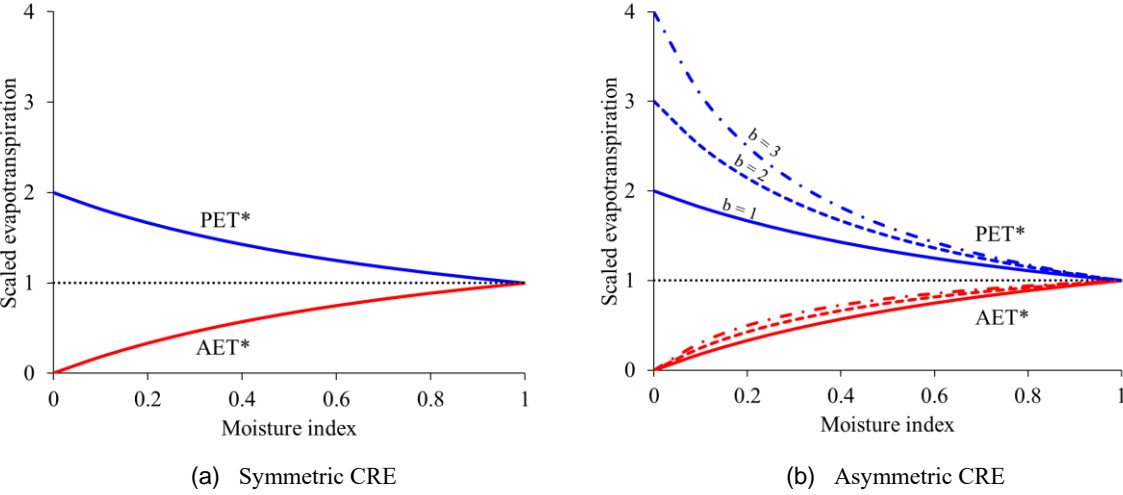

(a) Symmetric CRE        (b) Asymmetric CRE

**Figure 5.** Complementary relationship of evapotranspiration that reconstructed the relationship of Bouchet (1963), Hobbins et al. (2001a) and Kahler and Brutsaert (2006)

Based on this CRE hypothesis, Brutsaert and Parlange (1998) suggested that a decrease in $E_{pan}$ could signal an increase in AET. The observed data of the large-scale conterminous US (Lawrimore and Peterson, 2000; Hobbins et al., 2004; Walter et al., 2004) and Yellow River basin of China (Liu et al., 2006) and regional shorter timescale scale (Kahler and Brutsaert, 2006) have confirmed the CRE. However, there is still a few debates about the interactive mechanism among energy of drying power and different kinds of evapotranspirations. Some point out on the validity of the CRE hypothesis. Others point out the uncertainty of the observational data while acknowledging the validity of the hypothesis. Liu et al. (2004) attribute the $E_{pan}$ decline in China to decreasing solar irradiance, not to the CRE. Tebakari et al. (2005) found that the CRE was not applicable in the Chao Phraya River basin of Thailand using 27 stations for the period of 1982–2000. Regardless of the validity of the CRE hypothesis, it is generally accepted that accurate measurement of evapotranspiration is very difficult and the uncertainty is relatively high.

In the CRE equation, the PET and AET converges to WET as the atmospheric moisture index increases. The WET is determined by the available energy at the ground surface. When the available energy is constant and the surface is gradually dry, AET occurs less than WET, and PET gradually increases as the sensible heat flux increases. Moisture Availability (MA) was defined to reflect changes in AET and $E_{pan}$ (PET) at the same time. These indicators can explain the degree of approximation of the AET to the PET.

### 3.2 Estimation of PET

PET is defined as the environmental demand for evapotranspiration, i.e., the evapotranspiration rate of a specific crop or ecosystem if there were sufficient water available (Kahler & Brutsaert, 2006). The PET is a measure of demand side and would occur if a sufficient moisture source were available. Similarly, ET0 (reference crop evapotranspiration) is defined as the





evapotranspiration rate from a reference surface, a hypothetical grass reference crop with specific characteristics, under the condition of unlimited provision of water. Food and Agriculture Organization of United Nations (FAO) discourages the use of the denominations such as PET to ambiguities in their definitions (Allen et al., 1998). The problem is evapotranspiration
measurement data is limitedly available due to their high cost, complex installation, and/or intensive maintenance and the observation network is extremely limited even in developed countries. Therefore, $E_{pan}$, the evaporation from an open water surface providing an index of the integrated effect of radiation, air temperature, air humidity and wind on evapotranspiration, has been used widely by applying empirical coefficients to relate $E_{pan}$ to $ET_0$. $E_{pan}$ is advantageous in measuring and availability but is prone to too many sources of uncertainties to be used for trend analysis (Abtew et al., 2011). There is a number of ways
to estimating evaporation from free water surface, e.g. water and energy budget, Penman / Penman-Monteith equations and direct measurement from pan devices. They tried to be compared for lake modeling studies and the energy-budget method is the most reliable (Swancar et al., 2000; Rosenberry et al., 2007; Winter, 1981). The directly measured $E_{pan}$ generally gives overestimation because a pan is exposed to larger amount of solar radiation through its base and wall. Moreover, the gaps between the evaporations from a pan and a natural free water surface vary with the seasonal and annual differences in radiation,
air temperature, wind and heat storage within the water bodies. The commonly used pan coefficient is known to be 0.7 (Dingman, 2002). However, the recent water budget studies in Australia give 0.8 for the Tullaroop reservoir (Hagerty, 2006), 0.81 for lake Bolac (Raiber, 2008), 0.5-0.9 for Lakes Corangamite and Colac (Tweed et al., 2009) and 0.9 for Lake Wallace (Fawcett, 2005).

The only factors affecting PET or $ET_0$ are climatic parameters and can be computed from weather data. PET expresses the
evaporating power of the atmosphere at a specific location and time of the year and does not consider the crop characteristics and soil factors. The Penman equation (Penman, 1948) induces latent evaporation in the form of a convex linear combination of surface energy availability and evaporation rate due to water vapor transfer (Eq. [3]).

$$PET_P = \frac{\frac{\Delta}{\gamma}(R_n - G) + E_A}{\Delta + \gamma} \qquad [3]$$

where $PET_P$ is open water evaporation rate (kg/m2s), $\Delta$ is slope of saturation vapor pressure curve at air temperature (kPa/°C),
$R_n$ is net radiation (W/m2), $\gamma$ is psychrometric constant (kPa/°C), and $E_A$ is isothermal evaporation rate (kg/m2s). Later, Monteith (1965) proposed the Penman-Monteith equation, which improved the characteristics of the diffusion process by replacing the convective driver (wind function) of the Penman equation.

The Reference Evapotranspiration ($ET_o$) is defined as the rate of evaporation from a hypothetical reference surface under standardized meteorological conditions. The FAO has defined the reference surface as "a hypothetical reference crop with an
assumed crop height of 0.12 m, a fixed surface resistance of 70 sec/m, and an albedo of 0.23." This implies that the reference surface pertains to an extensive area of uniformly tall, actively growing green grass that completely shades the ground and has an adequate water supply. The FAO Penman-Monteith equation is widely recognized as the standard method for estimating $ET_o$ due to its comprehensive consideration of meteorological factors influencing evapotranspiration (Walter et al., 2000;





Droogers et al., 2002; Lage et al., 2003). The evapotranspiration rates of various crops can be estimated from the reference

surface evapotranspiration rate (ET$_0$) by employing crop coefficients. The Penman-Monteith equation integrates the driving

components of solar radiation, air temperature, humidity, wind speed, and atmospheric pressure to simulate the evaporation

that would occur from a well-watered, large, and homogeneous grass or reference crop surface (crop coefficient K$_o$=1). This

standardization allows for consistent comparison of ET$_o$ across different regions and climates.

Among the temperature-based methods, Hargreaves and Samani (1985) introduces a simple and widely used method for

estimating reference evapotranspiration (ET$_o$) using temperature data. The Hargreaves-Samani method correlates ET$_o$ with

daily temperature fluctuations, providing a practical alternative to more complex models for regions lacking extensive

meteorological data. In this study, the pan evaporation (E$_{pan}$) was used as the PET amount, and the observed E$_{pan}$ values were

corrected using the FAO P-M equation as described in Section 3.3. The FAO P-M equation, derived from the Penman-Monteith

equation, is presented in Equation [4] (Allen et al., 1998).


$$PET_{FAO\ P-M} = \frac{0.408\ \Delta(R_n - G) + \gamma \frac{900}{T+273}u_2(e_s - e_a)}{\Delta + \gamma(1 + 0.34u_2)} \qquad [4]$$

where PET$_{FAO\ P-M}$ is the Penman-Monteith reference evapotranspiration (mm/day), R$_n$ is the net radiant energy (MJ/m2/day)

that is purely accumulated on the surface (or crop, e$_s$ is the saturated vapor pressure (kPa), e$_a$ is the actual vapor pressure (kPa),

Δ is the slope of vapor pressure curve (kPa/°C), and γ is the humidity coefficient constant (kPa/°C). The difference between es

and e$_a$ is the moisture deficit (kPa). G is the soil heat flux density (kPa). The daily value can be ignored as it is relatively small

compared to other variables (Allen et al., 1998). The observed values were used for the e$_s$ and e$_a$, and the estimating process

for the remaining variables are described in the section 3.3.1 ~ 3.3.4.

### 3.2.1 Estimating net radiation energy

The net radiant energy (R$_n$) is calculated as the difference between the net short wave radiated energy (R$_{ns}$) and the net long

wave radiated energy (R$_{nl}$) (Eq. [5]).

$$R_n = R_{ns} - R_{nl} \qquad [5]$$

where $R_n$ is the daily net radiation energy (MJ/m2/day), $R_{ns}$ is the daily shortwave radiation energy (MJ/m2/day), and $R_{nl}$ is

the daily net long wave radiation energy (MJ/m2/day). As in the Eq. [6], $R_{ns}$ is the solar radiation energy excluding the

reflected amount.

$$R_{ns} = (1 - \alpha)R_s \qquad [6]$$

where α is the surface reflectance (albedo) and 0.23 was used in this study. $R_s$ is the solar radiation energy incident on the

Earth's surface (MJ/m2/day).



Among the meteorological stations around the Yongdam Dam basin, Jeonju is the only meteorological station providing observational data of solar radiation energy. Therefore, in this study, solar radiation energy was calculated using the Eq. [7]

and meteorological data instead of using observed data.

$$R_s = \left( a_s + b_s \frac{n}{N} \right) R_a \qquad [7]$$

where n is the actual daylight hours, N is the maximum possible daylight hours, $a_s$ and $b_s$ are Angstrom constants, and $R_a$ is the external radiant energy (MJ/m2/day) same as Eq. [8].

$$R_a = \frac{24 \times 60}{\pi} G_{sc} d_r [\omega_s \sin(\varphi) \sin(\delta) + \cos(\varphi) \cos(\delta) \sin(\omega_s)] \qquad [8]$$

where $G_{sc}$ is the solar constant (0.0820MJ/m2/day), φ is the latitude (radian), $d_r$ is the relative reciprocal of the distance between the sun and the earth, δ is the sun's declination (radian), and $\omega_s$ is the angle (radian) of the sun at the sunset time. The $d_r, \delta, \omega_s$ are estimated using the Eq. [9] ~ [11].

$$d_r = 1 + 0.033 \cos \left( \frac{2\pi}{365} J \right) \qquad [9]$$

$$\delta = 0.409 \cos \left( \frac{2\pi}{365} J - 1.39 \right) \qquad [10]$$

$$\omega_s = \arccos[-\tan(\varphi) \tan(\delta)] \qquad [11]$$

where J is the Julian date (J=1 on January 1, J=365 or 366 on December 31), φ is the latitude (radian), and δ is the sun's declination (radian).

Next, the net longwave radiation energy is estimated using the Eq. [12].

$$R_{nl} = \sigma \left[ \frac{T_{max,K}^4 + T_{min,K}^4}{2} \right] \left( 0.34 - 0.14 \sqrt{e_a} \right) \left( 1.35 \frac{R_s}{R_{so}} - 0.35 \right) \qquad [12]$$

where the $\sigma$ is $4.903 \times 10^{-9}$ MJ/K4/m2/day of Stefan-Boltzman constant. The $T_{max,K}$ is the maximum daily temperature(K), $T_{min,K}$ is minimum daily minimum temperature(K). The $e_a$ is actual vapor pressure(kPa), $R_s$ is solar radiation incident to the earth's surface, $R_{so}$ is the net solar radiation energy (MJ/m$^2$/day) on clear sunny day. The equation for estimating $e_a$ is expressed in the Eq. [13].

$$e_a = \frac{RH_{mean}}{100} \left[ \frac{e_0(T_{max}) + e_0(T_{min})}{2} \right] \qquad [13]$$

where RH$_{mean}$ means the average daily relative humidity (%). The $e_0(T_{max})$ is the saturated vapor pressure (kPa) at the maximum daily temperature, and $e_0(T_{min})$ is the saturated vapor pressure(kPa) at the minimum daily temperature. The $e_0(T)$ is calculated through the Eq. [14].





$$e_0(T) = 0.6108 \exp\left[\frac{17.27T}{T+237.3}\right] \qquad [14]$$

The $R_{so}$ is the net solar radiation energy on a clear sunny day without the influence of clouds or fog and it means the amount of radiation energy when the daylight hours and the daytime hours are the same (n/N=1) (Eq. [15]).

$$R_{so} = (a_s + b_s)R_a \qquad [15]$$

where 0.25 and 0.5 were used for the Angstrom constants $a_s$ and $b_s$ as derived by Allen et al. (1998). The external radiation energy $R_a$ is estimated using the Eq. [8].

$$u_2 = u_z \frac{4.87}{\ln(67.8\,z - 5.42)} \qquad [16]$$

where z is the distance from the ground to the observation site (m), and $u_z$ is the wind speed (m/s) at z m.

### 3.2.3 Slope of saturation vapor pressure curve

The slope of saturation vapor pressure curve, that is, the proportionality constant representing the instantaneous rate of the actual vapor pressure with respect to Celsius temperature ($T$) can be estimated as in Eq. [17].

$$\Delta = \frac{4098\left(0.6108 \exp\left(\frac{17.27T}{T+237.3}\right)\right)}{(T+237.3)^2} \qquad [17]$$

### 3.2.4 Estimation of psychrometric constant

The psychrometric constant is a value that applied to correct the error of humidity estimation using a psychrometry, and Eq. [18] is defined as an equation for atmospheric vapor pressure.

$$\gamma = 0.665 \times 10^{-3}\,P \qquad [18]$$

where P is the atmospheric vapor pressure (kPa). The FAO guideline presents the formula for P as a function of altitude (Eq. [19]).

$$P = 101.3 \times \left(\frac{293 - 0.0065z}{293}\right)^{5.26} \qquad [19]$$

where z is the elevation (m) of the psychrometer.

The net solar radiation and moisture deficit are the major driving components for the PET whereas the wind speed seems to be the minor component with high scatteredness.



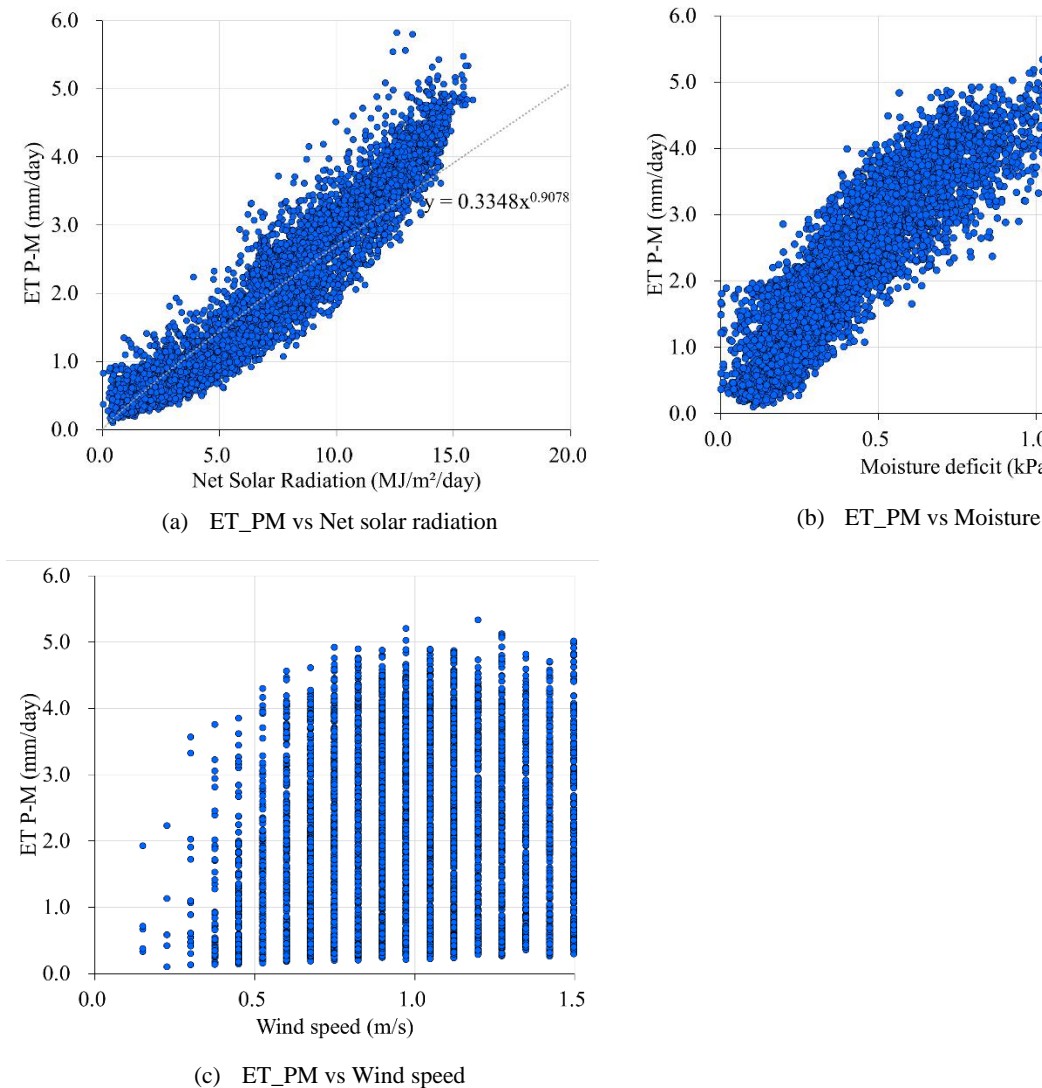

Figure 6 ET_PM vs Driving components

3.3 Estimation of wet environment evapotranspiration (WET)

In wet environments, evapotranspiration (ET) refers to the combined process of evaporation from surfaces and transpiration from vegetation, influenced by ample soil moisture and high atmospheric humidity. The Priestley and Taylor equation (Priestley and Taylor, 1972) provides a method to estimate potential evapotranspiration ($ET_o$) under these conditions. It is based on the assumption that in wet environments, vegetation actively regulates water loss through transpiration, and thus,

the evaporation component dominates. The equation calculates $ET_o$ as a function of net radiation ($R_n$) and the psychometric constant ($\gamma$), where a dimensionless parameter ($\alpha$) adjusts for the efficiency of evaporation in wet conditions compared to



potential evaporation in dry conditions. This approach assumes that in wet environments, available energy primarily drives evaporation, with transpiration contributing less significantly compared to drier conditions. Therefore, the Priestley and Taylor equation is particularly suited for estimating ET$_o$ in regions with abundant moisture availability, where transpiration is not

limited by water stress.

     Due to the root water uptake capacity of vegetation, transpiration is generally approximated by WET, which is estimated using the Priestley-Taylor equation. WET constitutes a fundamental component in various evapotranspiration (ET) models, including complementary relationship models (Aminzadeh and Or, 2017; Szilagyi et al., 2017), residue models (Bastiaanssen et al., 1998; Crago et al., 2016; Jia et al., 2012; Su, 2002), and triangle/trapezoid models (Garcia et al., 2014;

Jiang and Islam, 1999; Long and Singh, 2012), among others. Conducting controlled experiments to artificially saturate land surfaces on a regional scale poses significant challenges. The Priestley-Taylor equation, in conjunction with the Penman equation, has been extensively applied to validate the complementary relationship between potential evapotranspiration (PET) and actual evapotranspiration (AET). There are numerous methods for estimating PET, AET and WET and each method has the advantages and limitations (Brutsaert and Stricker, 1979; Morton, 1983; Hobbins et al., 2001). The choice between them

depends on available data and the specific environmental characteristics of the study area. The topic is explored in detail in the subsequent section.

The P-T equation is a simplified form of the Penman equation as one of the methods using the solar radiation (Eq. [20]).

$$ET_{P-T} = \alpha \frac{\Delta(R_n - G)}{\Delta + \gamma} \qquad [20]$$

where $ET_{P-T}$ is the WET estimated using the Priestly-Taylor equation. The $\alpha$ is the Priestley-Taylor coefficient. The $R_n$ is

the net radiation energy (MJ/m$^2$/day). The G is the soil heat flux density (MJ/m$^2$/day). The $\Delta$ is the slope (kPa/℃) of the saturation vapor pressure curve. The $\gamma$ is the psychrometric constant (kPa/℃).

Comparing the Penman equation with the Priestley-Taylor equation indicates that the aerodynamic term of the Penman equation was replaced with the coefficient α in the Priestley-Taylor equation. The WET is lower bound of PET assuming the ambient air is fully saturated and it shows seasonal oscillations of upper and lower limits as well, and shows high scattering

only in summer (Figure 7). The constant value of 1.26 significantly underestimates the PET under arid and semi-arid climate conditions (Ai and Yang 2016). The Penman-Monteith ET(PET) is the weighted sum of radiation-driven ET and aerodynamics ET and the Priestley-Taylor ET(WET) is solely proportional to the radiation-driven ET. The proportionality of PET and WET to net solar radiation is comparative in the sense that the WET shows a narrow and clear bounds especially upper bound (Figure 6 & 8).






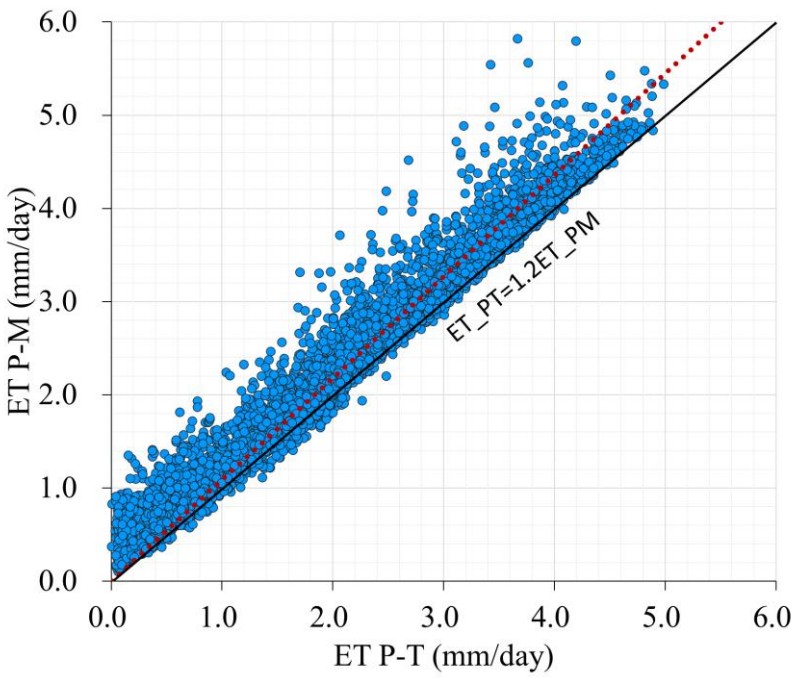

**Figure 7** PET w.r.t WET, the lower bound of PET assuming the ambient air is fully saturated (Jangsu station)

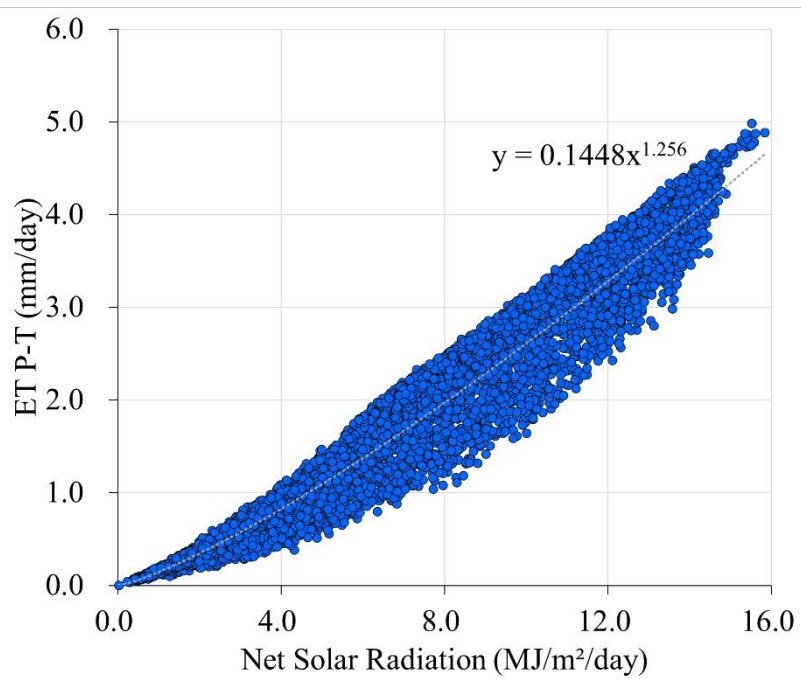

**Figure 8** ET_P-T w.r.t. net solar radiation






## 4 Results

4.1 Correction for pan evaporation

In the Stainless-Steel Class A evaporation pan, the evaporation tends to be overestimated under the influence of the surrounding atmospheric and environmental conditions, so the pan evaporation is corrected by applying a pan correction factor

(kp). The kp is calculated as the average value of the slope of the monthly standard evaporation and monthly pan evaporation calculated using the FAO P-M equation. Figure 9 shows the correlation between the FAO Penman-Monteith's PET and Epan, and Figure 9(a) indicates that the correction factor for the Yongdam Dam basin is 0.768. The result of applying the correction factor to Epan is shown in Figure 9(b).

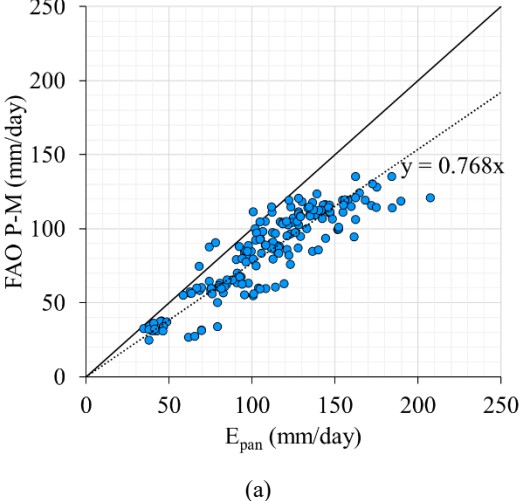

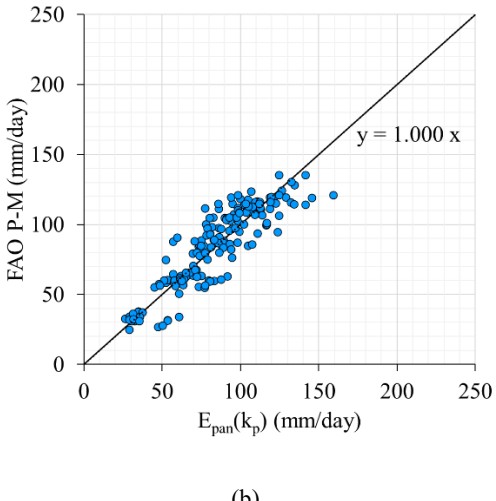

(a)                                                        (b)

**Figure 9.** The scatter plot between the FAO P-M's PET and $E_{pan}$ (a) Original $E_{pan}$ observation data, (b) Corrected $E_{pan}$ using the correction
factor

4.2 Estimation of WET under CRE model

The evapotranspiration complementary relationship represents the relationship between the dimensionless PET and the AET with respect to Moisture Availability (MA) index. The MA index is an indicator of the degree of saturation of the land surface and uses the ratio of the AET and the PET. In other words, the MA is 0, it means that the land surface is dry, and

if it is close to 1, the AET approaches the PET meaning that the land surface is close to saturated state. In order to make the PET and AET non-dimensional, each is divided by the WET. Under the complementary relationship, when the MA is close to 0, the dimensionless PET (PET*) approaches the maximum value, and the dimensionless AET (AET*) approaches the minimum value, 0. When the MA is close to 1, PET* and AET* is close to each other and approaches to 1. In other words, when the land surface and the ambient air are saturated, the PET, AET, and WET all have the same value. Here, the important

condition for the complementary relationship being validated is that the land surface and the ambient air must have the positive




correlation with each other. In other words, when the land surface is dry and AET decreases, then the atmosphere must also be dry, then the PET is maximized. Meanwhile when the land surface is saturated and AET is also maximized, then the atmosphere also becomes saturated, the PET is minimized and becomes equal to WET (Figure 5). However, in reality, the actual vapor pressure of the atmosphere within the Yongdam Dam basin forms the minimum value of the evaporation flux

occurring on the land surface and shows the tendency of completely random distribution. In other words, the humidity of ambient air is formed as a result of the amount of moisture in the air mass flowing in from the outside, and this shows little correlation with the degree of saturation of land surface. As the MA increases, the actual vapor pressure becomes equal to the saturation vapor pressure, and the moisture deficit decreases and close to 0, but in the Yongdam Dam basin, there is no correlation between the MA and the moisture deficit (Figure 11). At the same time, WET, which is used for transforming PET

and AET into nondimensional variables, also shows a random distribution regardless of MA (Figure 12).

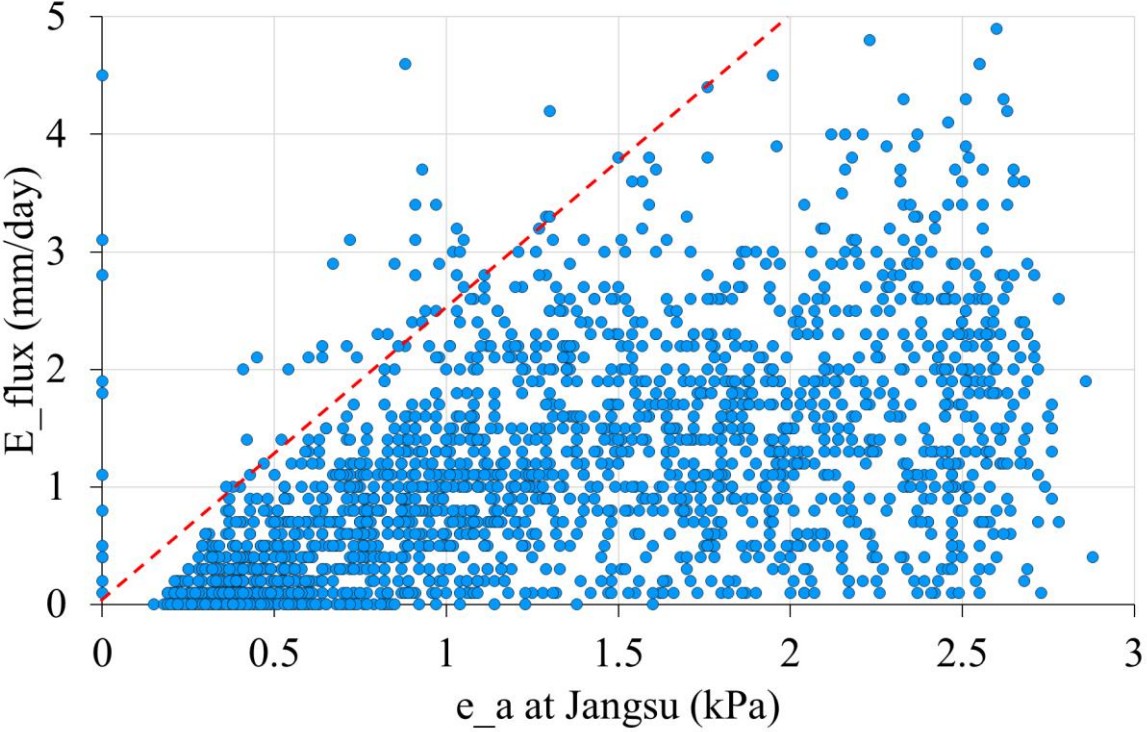

**Figure 10** Scatteredness between ET flux and atmospheric vapor pressure



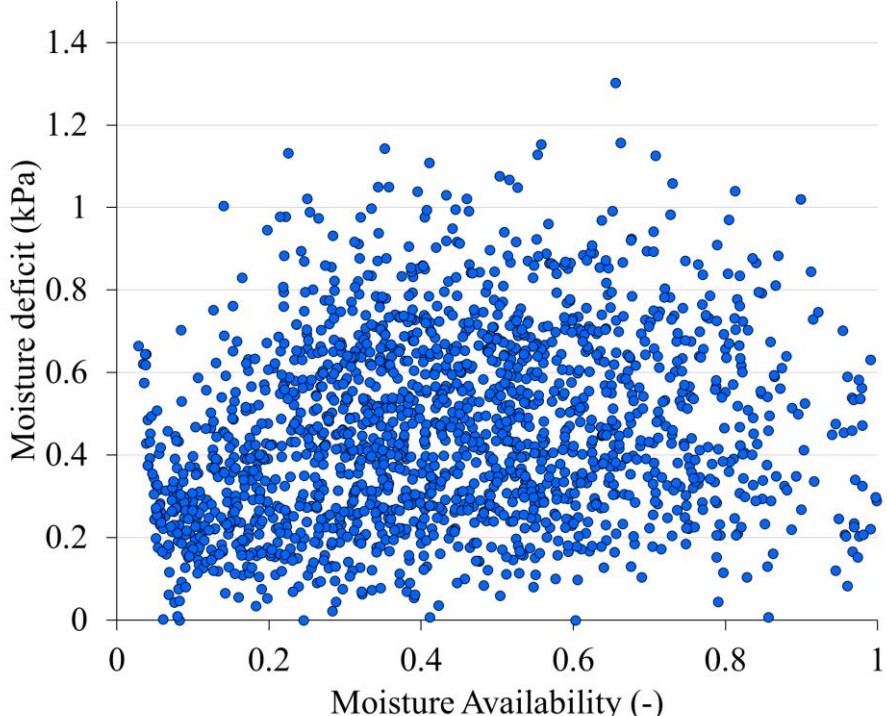

**Figure 11** Scatteredness between Moisture Availability(MA) and Moisture Deficit(MD)



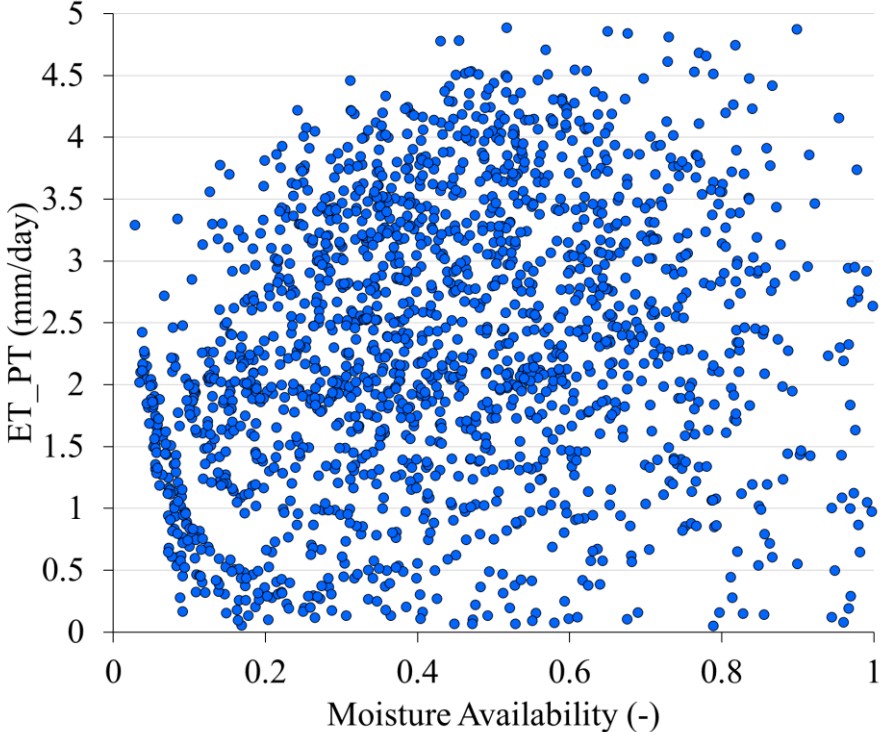

**Figure 12** Scatteredness between Moisture Availability(MA) and WET

### 4.2 Complementary relationship of ET in Yongdam dam basin

To confirm the validity of the Bouchet's CRE curve under the real observation data in the Yongdam dam basin, the three different kinds of daily ET data during 2011 to 2019 were used; namely the AET observed from flux tower, the PET based on the reference ET proposed by FAO P-M equation, and the WET from the Priestley-Taylor equation. As a results of complex correlation structures among PET, AET, WET and MA, the complementary relationship is shown in Figure 13. The lower bound of AET* is linearly proportional to MA which can be expected from the pattern between PET and WET seen in Figure 7 unlike the typical parabolic pattern of AET* in Figure 5. The PET* was slightly downward linearly for increasing MA but distributed mostly around 1 for entire range of MA, very different from the typical complementary relationship line, and tended to be somewhat scattered for low MA. This might be the result of the scatteredness between MA and moisture deficit (Figure 11), and MA and WET (Figure 12). The relationship between PET and dimensionless PET(PET*) disregarding the effects of AET shows the pattern more clearly (Figure 13). In other words, the relative variability of PET with respect to WET tends to decrease along with increasing PET (Figure 7 and 14).



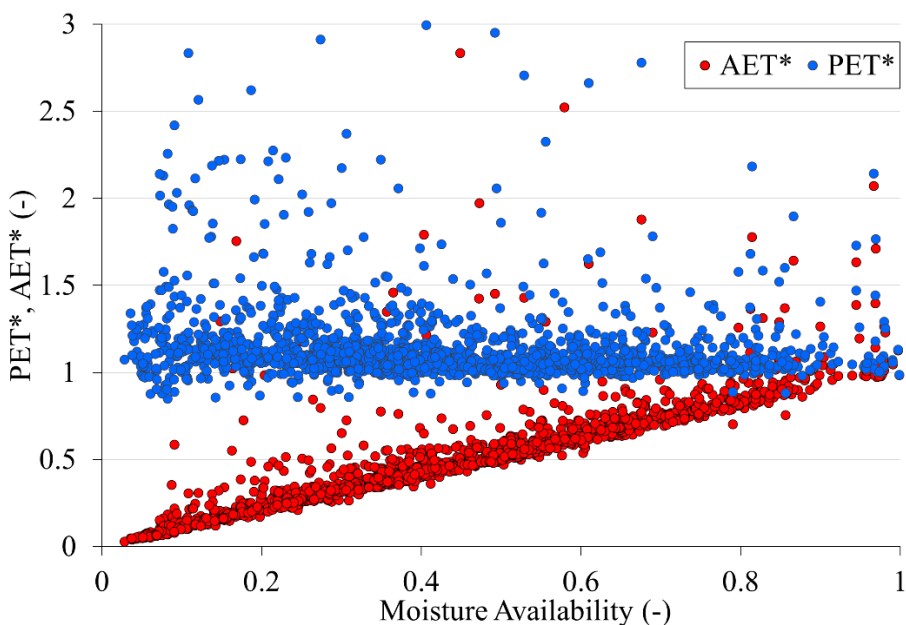

**Figure 13** The complementary relationship among ETs and MA

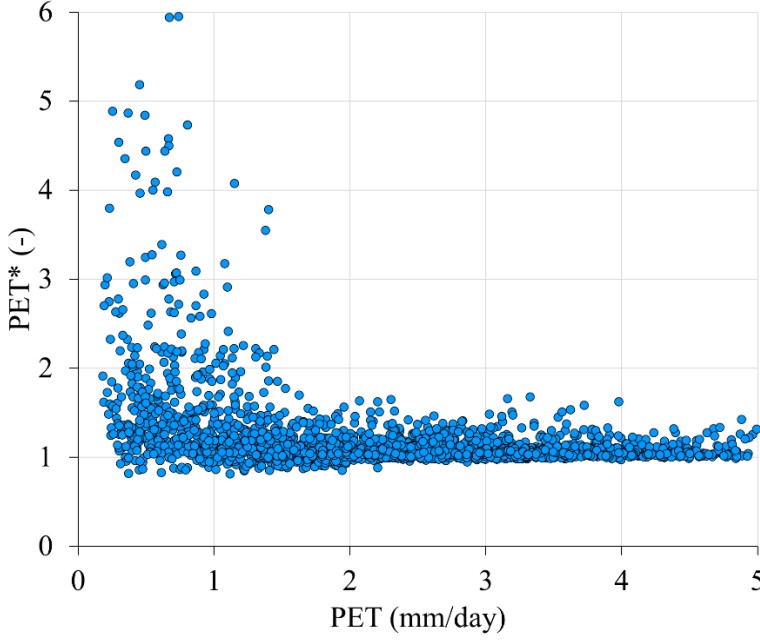


**Figure 14** Relative variability between PET and dimensionless PET(PET*)





As mentioned, the MA in the complementary relationship is the degree of saturation at land surface and it is not correlated with the vapor pressure at the Yongdam dam basin. Now AET* and PET* are plotted again with respect to the relative humidity of the ambient air. The AET* is highly scattered (Figure 15(a)). The PET* has distinct lower bound and the scatteredness

decreases along with increasing RH.

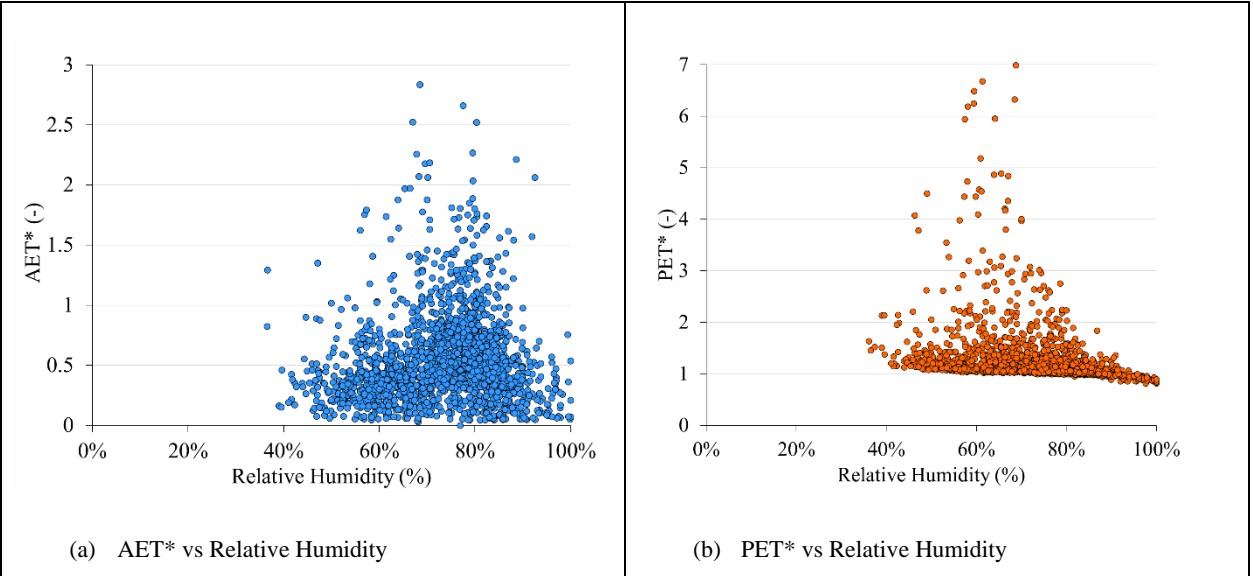

(a)   AET* vs Relative Humidity          (b)   PET* vs Relative Humidity

**Figure 15** AET* and PET* with respect to relative humidity

## 5 Conclusion

Since the introduction of the complementary relationship theory and its subsequent development, numerous studies have been conducted to validate the proposed hypothesis or to predict actual evapotranspiration based on the theory. Despite these efforts, the substantial variability inherent in the data used to elucidate the relationship has often obscured the underlying trend. Notably, the application of the complementary relationship of evapotranspiration to the Korean Peninsula region, situated within a typical monsoon climate area, has not been previously documented. However, the outcomes of this study

unveil a noteworthy departure of the complementary relationship in this region from the anticipated norm. The pivotal factor contributing to this divergence is the unexpected absence of correlation between ground moisture content and atmospheric humidity, challenging the conventional belief and fundamental assumption of a robust positive correlation between these two components.

         The conventional form of the complementary relationship of evapotranspiration (ET) posits that potential

evapotranspiration (PET*) diminishes, while actual evapotranspiration (AET*) increases in response to the moisture



availability of the land surface. The resulting relationship curve exhibits a vertically symmetric pattern cantered around the WET. Subsequently, the asymmetric formulation of the complementary relationship employing the psychrometric constant (b) and the slope of the saturation vapor pressure was introduced. Various studies on asymmetric complementary relationships have since emerged. In the context of the dynamically active seasonal climate prevalent in the Korean Peninsula, the symmetric

complementary relationship loses its validity under conditions of pronounced meteorological fluctuations, such as those encountered in the monsoon climate environment of Korea. When explicating this deviation using asymmetric complementary relationships, it is imperative to note that the psychrometric constant (b) assumes values significantly below 1 and approaches zero, signifying the convergence of PET* toward the WET. In this paper, it has been verified that PET* does not exhibit a proportional response to moisture availability, with WET serving as the minimum boundary of PET*.

This paper presents the outcomes of an analysis utilizing Evapotranspiration (ET) flux observations and weather station data situated in the Yongdam Dam basin within the Geum River basin, South Korea. Despite the fact that this geographical area falls within the monsoon climate zone and displays conspicuous seasonal variations, premature generalization of the findings as representative of the entire monsoon region is cautioned. Variations exist even within the monsoon climate zone, where coastal and inland climates, as well as distinctions in flat, mountainous, and bowl-shape basin

terrains surrounded by mountains, contribute to disparate climatic characteristics. Notably, the study verifies the presence of complementary relationship characteristics in regions demonstrating strong correlations between soil moisture and air humidity, such as deserts and tropical areas, where annual climate fluctuations and seasonal winds exhibit less significance. It is essential to recognize, however, that the correlation between soil moisture and air humidity weakens in regions where externally introduced factors, such as the dominant influence of the monsoon climate zone, impact the climate. In such cases, potential

evaporation and actual evaporation tend to deviate from the anticipated complementary relationship and assume a more random distribution. This consideration is pivotal when endeavouring to estimate actual evaporation based on the complementary relationship, particularly in scenarios where external factors, such as the influence of the monsoon climate zone, lead to a departure of potential and actual evaporation from the expected pattern.

**Competing interests**

The contact author has declared that none of the authors has any competing interests.

**Acknowledgments**

This work was supported by the National Research Foundation of Korea (NRF- 2021K1A3A1A20003375).



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
