# Peer review of "Evapotranspiration Dynamics in Monsoon-dominated Region in the Korean Peninsula"

_EGUsphere, 2024_

## Referee Comment (RC1)

The submitted paper "Evaporation Dynamics in Monsoon-Dominated Regions of the Korean Peninsula", by Kim, et.al, attempts to demonstrate that the complementary relationship of Evapotranspiration (CRE), does not hold true for monsoon-dominated region. A fundamental condition for CRE to be validated is that moisture availability in the land surface and the ambient air, contributing to actual evapotranspiration (AET) and potential evapotranspiration (PET) respectively, should be correlated. This correlation exists in regions where the impact of advection is minimal. However, in monsoon-dominated regions, the external moisture input into the atmosphere disrupts the moisture dynamics between the land surface and the atmosphere, thus invalidating CRE.

This point is effectively highlighted in Figure 10, which is the figure I found commendable in the article. It shows that actual evapotranspiration (ETa) from the flux tower does not correlate with the vapor pressure deficit in the atmosphere, thereby disproving CRE in monsoon-dominated regions. While the violation of CRE in regions with significant advection can be inferred from the derivation of CRE, the novelty of Figure 10 lies in using evapotranspiration data, calculated from the flux tower. This provides a more direct measure of ETa compared to previous studies that relied on ETa, estimated from the water balance models at a regional scale.

However, apart from this, the research paper is poorly written. The figures are not well-conceived, the text is confusing regarding the use of different evapotranspiration (ETs), and numerous other issues are present. Therefore, I suggest to the editor that this paper is not currently suitable for acceptance in this reputed journal. I will highlight the major concerns point by point below:

1) The abstract lacks clarity and specificity regarding the major research findings.

   a) For instance, the statement, "*This study investigates the dynamics of evapotranspiration in a monsoon-dominated region of the Korean Peninsula, focusing on the challenges associated with measurement, identification, and prediction of potential and actual evapotranspiration,*" **does not align with the paper's content, as the discussion on these challenges are not evident throughout the whole document**.

   b) Additionally, the claim**,** *"This research confirms the existence of complementary relationship behavior in regions with strong correlations between soil moisture and air humidity, such as deserts and tropical areas,"* **is misleading because the study is exclusively focused on the Korean Peninsula only.**

2) The introduction is lengthy and detailed, obscuring the core reason for the paper's necessity.

3) The purpose of each figure in the article is unclear, and their contribution to the overall conclusion is not well-articulated. It appears that the author included all figures without considering their specific relevance or how they support the main conclusions.

4) The text is unclear about the use of PET and WET. It appears that potential evapotranspiration (PET) was calculated using the Penman-Monteith method, and wet evapotranspiration (WET) was derived from the Priestley-Taylor (P-T) equation. However, the role of the pan evapotranspiration data collected at Jeonju is ambiguous, as it is only mentioned in the context of calculating the pan evaporation coefficient.

A significant issue is the choice not to use pan evaporation (EPan) as the measure of PET, as done in studies like Ramirez et al. (2005), which is also referenced in this study. Using EPan as PET would establish it as the upper limit of evapotranspiration in the region. Currently, WET from the P-T method is acting as the maximum limit. While this might not be the major issue as we can still interpret the results, but the author needs to warrant all the rationale behind such selection and usage also highlighting how (if) it impacts the analysis.

**5) Regarding the main highlight of the paper, Figure 13, there are several concerns:**

a) It appears that each dot represents a daily observation. Most studies on the complementary relationship of evapotranspiration (CRE) use annual scale observations for different basins. The choice to use a daily scale needs clarification. The introduction should specify whether the author intends to validate CRE on a daily scale.

b) Typically, the x-axis in the CRE hypothesis or Budyko framework reflects long-term water availability, as it represents climatic conditions and aids in predicting actual evapotranspiration (AET) based on the region's characteristics. This is usually represented by the ratio of long term mean annual precipitation (P) to PET (Budyko) or the potential humidity index (phi) as the ratio of annual precipitation to WET (Ramirez et al., 2005). The author, however, calculates the moisture availability index as the ratio of AET to PET (lines 186-187) in daily scale (as I understood). The rationale for this choice should be explained (also related to point a).

c) To better understand how AET and PET change with moisture availability, it may be necessary to include at least one additional variable in computing moisture availability, rather than current formulation of moisture availability (MA) as the ratio of AET to PET. Relying solely on calculating MA from the ratio of AET to PET and trying to explain the dynamics of AET and PET based on that ratio might

obscure the relationship between AET and moisture availability (MA). For instance:

(i) The blue line in the graph indicates that PET/WET (y axis) is nearly equal to 1 for all AET/PET values (moisture availability), which suggests that PET is approximately equal to WET across different moisture conditions.

(ii) The red line shows that AET/WET is almost linearly proportional to AET/PET (moisture availability), implying that PET is proportional to WET, unless the temporal scales of x-axis and y-axis are different (similar to (i)). This approach obscures how AET responds to moisture availability. Therefore, the x-axis could represent long-term moisture availability or catchment characteristics more effectively, and alternative methods of representation of moisture availability should be considered.

d) Since the author utilizes AET from the flux tower as the measure of regional actual evapotranspiration, it is essential to highlight the potential fetch area of the flux tower, as it depends on the wind direction and speed. In the map, the flux tower appears to be located quite close to the dam. If the scales on the figure are accurate, it needs to be clarified whether the flux tower measures ET from the dam or the surrounding vegetation (forest) most of the time.

Apart from the major concerns, there are several minor issues:

1. The detailed explanation of how PT and PM evapotranspiration are calculated in Section 3 could be omitted unless there are deviations from standard practice. Given the extensive literature on these calculations, this section might be unnecessary unless it introduces novel methods or significant deviations.
2. The author should be more careful in sentence construction throughout the paper. For example:
   - Line 66-68: The sentence, "However, their study was constrained by the use of historical observational data, which may contain measurement inaccuracies and spatial variability, and the applicability of Bouchet's hypothesis to diverse climatic conditions beyond the studied regions remains uncertain without further validation," lacks clarity. **It is not clear how the current research addresses these constraints, particularly regarding observational data inaccuracies, uncertainties, or spatial variability.**
   - Line 455: The statement, "Notably, the study verifies the presence of complementary relationship characteristics in regions demonstrating strong correlations between soil moisture and air humidity, such as deserts and tropical areas, where annual climate fluctuations and seasonal winds exhibit less significance," is confusing. **Since the study focuses only on the Korean Peninsula, it is unclear how it verifies CRE in deserts and tropical areas. If the author intends to reference existing literature, this should be explicitly stated and clearly articulated.**

---

## Author Comment (AC1)

**Answer for the comments**

Thank you for your insightful comments. Here is our detailed response:

Among the major comments from the reviewer, I would like to address the methodology and conclusion sections. The related comments are as follows:

1) The abstract lacks clarity and specificity regarding the major research findings.

a) For instance, the statement, "This study investigates the dynamics of evapotranspiration in a monsoon-dominated region of the Korean Peninsula, focusing on the challenges associated with measurement, identification, and prediction of potential and actual evapotranspiration," does not align with the paper's content, as the discussion on these challenges are not evident throughout the whole document.

**Answer)** The complementary relationship hypothesis regarding evapotranspiration, first proposed by Bouchet (1963) and Budyko (1974), has played an important role in explaining the evapotranspiration process. Many researchers have demonstrated the validity of this hypothesis in various climatic environments. The basic logic of this hypothesis is simple and clear: Initially, when the surface is dry and the atmosphere is also dry, potential evapotranspiration is at its maximum and actual evapotranspiration does not occur. As moisture on the surface increases, actual evapotranspiration begins, increasing atmospheric humidity and reducing the moisture gradient between the surface and the atmosphere, which in turn decreases potential evapotranspiration. When the surface moisture continues to increase to near saturation, the atmosphere also approaches saturation, and potential evapotranspiration and actual evapotranspiration converge to maintain a constant value (wet-environment evapotranspiration). While the natural behavior of evapotranspiration is straightforward, proving the complementary relationship hypothesis with actual measured data is challenging. This is because the hypothesis assumes a local situation where the inflow or outflow of air masses with different humidity levels from the outside is blocked, which is difficult to find in natural conditions. If there have been lengthy and confusing descriptions in the process of explaining these conditions and limitations of application, as well as the different conditions under which various studies have been conducted, these should be corrected in future updates of the document.

b) Additionally, the claim, "This research confirms the existence of complementary relationship behavior in regions with strong correlations between soil moisture and air humidity, such as deserts and tropical areas," is misleading because the study is exclusively focused on the Korean Peninsula only.

**Answer)** As previously explained, the environment most suitable for proving the complementary relationship hypothesis is one where there is a high correlation between soil moisture and atmospheric humidity. Cases where the soil is dry but the atmospheric humidity is high, or where the soil is wet but the atmosphere is dry, involve the inflow or outflow of air masses with different moisture contents due to lateral atmospheric motion (advection). While the local complementary relationship characteristics are inherent, they are mixed with external factors, making it difficult to distinguish them in the data. In the case of the Korean Peninsula, the synoptic-scale atmospheric motion is predominantly characterized by the inflow of warm and humid Pacific air masses into the inland areas. Although typhoons or localized rainfall can temporarily obscure these macro-scale atmospheric motions, the Pacific air mass generally dominates the monsoon climate in Korean peninsula. Therefore, during prolonged droughts, the atmosphere may be hot and humid while the soil remains dry, and during the rainy season, the soil may be wet while the atmosphere is dry, which hinders the CR appears clearly. This paper aims to assert that the complementary relationship may not be distinctly observed depending on the climate and that regions with strong seasonal advection are more likely to deviate from the complementary relationship.

2) The introduction is lengthy and detailed, obscuring the core reason for the paper's necessity.

**Answer)** This paper aims to assert that the complementary relationship may not be distinctly observed depending on the climate and that regions with strong seasonal advection are more likely to deviate from the complementary relationship. We agree with the comment that a concise edit focusing on the main argument is necessary.

3) The purpose of each figure in the article is unclear, and their contribution to the overall conclusion is not well-articulated. It appears that the author included all figures without considering their specific relevance or how they support the main conclusions.

**Answer)** We plan to revise the manuscript to effectively explain the connection between each figure and the main topic.

4) The text is unclear about the use of PET and WET. It appears that potential evapotranspiration (PET) was calculated using the Penman-Monteith method, and wet evapotranspiration (WET) was derived from the Priestley-Taylor (P-T) equation. However, the role of the pan evapotranspiration data collected at Jeonju is ambiguous, as it is only mentioned in the context of calculating the pan evaporation coefficient. A significant issue is the choice not to use pan evaporation (EPan) as the measure of PET, as done in studies like Ramirez et al. (2005), which is also referenced in this study. Using EPan as PET would establish it as the upper limit of evapotranspiration in the region. Currently, WET from the P-T method is acting as the maximum limit. While this might not be the major issue as we can still interpret the results, but the author needs to warrant all the rationale behind such selection and usage also highlighting how (if) it impacts the analysis.

**Answer)** WET stands for "Wet Environment Evapotranspiration". This term refers to the evapotranspiration that occurs under conditions where water for evapotranspiration is provided unlimitedly, essentially representing the maximum possible evapotranspiration given the available energy and environmental conditions. WET is commonly estimated by the Priestley-Taylor equation. Both PET and WET define the maximum limitation. However, PET is the maximum under given temperature and humidity and WET is the maximum under the unlimited moisture provision even when the air is fully saturated. PET can be estimated by FAO Penman-Montheith equation or directly by pan measurement. We used pan measurements from 3 national weather stations, one(Jangsu) inside the basin and 2(Jeonju and Geumsan) outside the basin. Even though Jeonju and Geomsan stations are outside the basin, they are located closest from the Yongdam dam site and the spatial variability of the pan evaporation was assumed not significant relatively. We used the FAO P-M for calibrating pan coefficient.

5) Regarding the main highlight of the paper, Figure 13, there are several concerns:

a) It appears that each dot represents a daily observation. Most studies on the complementary relationship of evapotranspiration (CRE) use annual scale observations for different basins. The choice to use a daily scale needs clarification. The introduction should specify whether the author intends to validate CRE on a daily scale.

**Answer)** As the reviewer indicated a number of CRAE paper have dealt with annual scale to capture synoptic scale behavior. However, recently there are comments on the needs for finer scale analysis. For example, Tu et al.(2023)* commented that their study acknowledged uncertainties in the estimation of Epa_max and the challenges of applying the CR model at shorter time scales and suggested further research is needed to test the model performance at daily or sub-daily scales.
* Tu, Z., Yang, Y., Roderick, M. L., & McVicar, T. R. (2023). Potential evaporation and the complementary relationship. Water Resources Research, 59, e2022WR033763. https://doi. org/10.1029/2022WR033763

b) Typically, the x-axis in the CRE hypothesis or Budyko framework reflects longterm water availability, as it represents climatic conditions and aids in predicting actual evapotranspiration (AET) based on the

region's characteristics. This is usually represented by the ratio of long term mean annual precipitation (P) to PET (Budyko) or the potential humidity index (phi) as the ratio of annual precipitation to WET (Ramirez et al., 2005). The author, however, calculates the moisture availability index as the ratio of AET to PET (lines 186-187) in daily scale (as I understood). The rationale for this choice should be explained (also related to point a).

**Answer)** The rationale for using the moisture availability index as the ratio of Actual Evapotranspiration (AET) to Potential Evapotranspiration (PET) on the x-axis of the Complementary Relationship (CR) graph lies in its ability to represent the degree of water limitation and the evaporative environment's condition. The ratio AET/PET directly indicates the availability of water for evaporation in a given environment. When AET is close to PET, it suggests that the environment is not water-limited, and the surface has sufficient moisture for evaporation to occur at its potential rate. Conversely, when AET is much lower than PET, it indicates water scarcity, as the actual evaporation is limited by the lack of available moisture. AET is most relevant variable for representing the moisture availability regardless of temporal scales. However, Budyko and Ramirez et al. (2005) used annual precipitation instead of AET because AET measured by flux sensor was guessed so limited at that time and the actual annual precipitation was best available component as an alternative for the AET. The recent paper on the CR (used AET measured by the flux tower and used the moisture availability as the ratio of AET/PET (Tu et al., 2023).

c) To better understand how AET and PET change with moisture availability, it may be necessary to include at least one additional variable in computing moisture availability, rather than current formulation of moisture availability (MA) as the ratio of AET to PET. Relying solely on calculating MA from the ratio of AET to PET and trying to explain the dynamics of AET and PET based on that ratio might obscure the relationship between AET and moisture availability (MA). For instance:
(i) The blue line in the graph indicates that PET/WET (y axis) is nearly equal to 1 for all AET/PET values (moisture availability), which suggests that PET is approximately equal to WET across different moisture conditions.
(ii) The red line shows that AET/WET is almost linearly proportional to AET/PET (moisture availability), implying that PET is proportional to WET, unless the temporal scales of x-axis and y-axis are different (similar to (i)). This approach obscures how AET responds to moisture availability. Therefore, the x-axis could represent long-term moisture availability or catchment characteristics more effectively, and alternative methods of representation of moisture availability should be considered.

**Answer)** The reviewer indicates the PET/WET ratio appears nearly 1 which means PET is approximately equal to WET regardless of moisture availability. Accurately saying, the PET*(=PET/WET) is mainly and randomly distributed between 1 and 1.2~1.3 and lower limit seems clearly 1. PET* is distributed above 1.2~1.3 which occurs when air is dry (PET>>WET). Usually in Korean peninsula during the most summer monsoon season period, humidity rises up to over 90%. Because the AET* is defined as the ratio of AET/WET and MA is defined as AET/PET, AET* vs MA shows distribution nearly above 1:1 line by its definition.

d) Since the author utilizes AET from the flux tower as the measure of regional actual evapotranspiration, it is essential to highlight the potential fetch area of the flux tower, as it depends on the wind direction and speed. In the map, the flux tower appears to be located quite close to the dam. If the scales on the figure are accurate, it needs to be clarified whether the flux tower measures ET from the dam or the surrounding vegetation (forest) most of the time.

**Answer)** The flux tower data are very limited in general. There are only 2 flux towers installed in the Yongdam dam basin, one in the Mt. Deogyu and the other one near the Yongdam dam. However, Yongdam dam flux tower was installed in 2017 so the recorded data are relatively short, and the sensor

is currently not available due to repair. Currently the data from the tower on the Mt. Deogyu are only available ones. As commented, we consider the effective coverage for a single flux tower and there will be uncertainties or biases in representing basin-wide areal measurements due to the spatial remoteness among individual measurements. Those aspects would be limitations of this work which should be improved as a lot more data will be accumulated.

We appreciate the opportunity to address these points and hope our response clarifies the rationale and methodology of our study.

---

## Author Comment (AC2)

Review comments (Vitali Diaz) and answers

Thank you for your insightful comments. Here is our detailed response:

Major comments

First, the title does not reflect what is presented in the manuscript; the authors present an analysis of the complementary relationship of the Evapotranspiration hypothesis in a basin on the Korean peninsula. A more appropriate title is

'Limitations of the complementary relationship of evapotranspiration hypothesis in a monsoon-dominated region in the Korean Peninsula: case study Yongdam basin'

**Answer)** We agree with the reviewer's suggestion.

Do the authors consider it necessary to further analyze the relationship above when this hypothesis does not perform well in areas such as the case study? Why do you venture to carry out this study? Which alternative, besides using this hypothesis, do you plan to perform?

**Answer)** [**Necessity of Further Analysis**] The complementary relationship hypothesis regarding evapotranspiration, first proposed by Bouchet (1963) and Budyko (1974), has played an important role in explaining the evapotranspiration process. Many researchers have demonstrated the validity of this hypothesis in various climatic environments. The basic logic of this hypothesis is simple and clear: Initially, when the surface is dry and the atmosphere is also dry, potential evapotranspiration is at its maximum and actual evapotranspiration does not occur. As moisture on the surface increases, actual evapotranspiration begins, increasing atmospheric humidity and reducing the moisture gradient between the surface and the atmosphere, which in turn decreases potential evapotranspiration. When the surface moisture continues to increase to near saturation, the atmosphere also approaches saturation, and potential evapotranspiration and actual evapotranspiration converge to maintain a constant value (wet-environment evapotranspiration). While the natural behavior of evapotranspiration is straightforward, proving the complementary relationship hypothesis with actual measured data is challenging. This is because the hypothesis assumes a local situation where the inflow or outflow of air masses with different humidity levels from the outside is blocked, which is difficult to find in natural conditions. If there have been lengthy and confusing descriptions in the process of explaining these conditions and limitations of application, as well as the different conditions under which various studies have been conducted, these should be corrected in future updates of the document.

Yes, we consider it essential to further analyze the relationship described by the Complementary Relationship of Evapotranspiration (CRE) hypothesis, even though it did not perform optimally in the Yongdam Dam basin case study. The main reason is that understanding the limitations and applicability of the CRE hypothesis under different climatic conditions, particularly in monsoon-dominated regions, is crucial. The observed deviations in our case study highlight the need to explore the interactions between potential evaporation (PET) and actual evaporation (AET) more deeply. This analysis will help in refining the hypothesis, improving its predictive capabilities, and providing more accurate tools for water resource management in complex climates like that of the Korean Peninsula.

[**Rationale for Conducting the Study**] Our venture to carry out this study stems from the significant role evapotranspiration plays in the hydrological cycle and water resource management. The CRE hypothesis has been validated in regions with stable climates, but its performance in monsoon areas

where strong seasonal advection dominates remains uncertain. By investigating its applicability in the Yongdam Dam basin, we aim to fill this gap and enhance our understanding of evapotranspiration dynamics in such regions. The study seeks to identify specific conditions under which the CRE hypothesis holds or is hidden without being exposed, thus contributing to the broader scientific discourse on evapotranspiration and improving climate and hydrological models for monsoon-influenced areas.

[Alternative Approaches] Our findings in this study are that the hot and humid air mass incoming from the Pacific ocean during the summer monsoon season governs the makes the PET(Potential ET) nearly equivalent to WET(Wet environment ET) regardless of moisture availability on land surfaces (Figure 13). The CR relationship is obscured and not clearly manifested due to the influx of hot and humid air masses from the Pacific. The research will be necessary for the evaporative dynamics under the environments governed by the advections by hot and humid air masses, which is required for exact prediction for losses from the reservoir water surfaces.

The time resolution is not very detailed when presenting the data and results, please be more explicit.

Answer) A number of CRAE paper have dealt with annual scale to capture synoptic scale behavior. However, recently there are comments on the needs for finer scale analysis. For example, Tu et al.(2023)* commented that their study acknowledged uncertainties in the estimation of Epa_max and the challenges of applying the CR model at shorter time scales and suggested further research is needed to test the model performance at daily or sub-daily scales. The daily data were used in this study.
* Tu, Z., Yang, Y., Roderick, M. L., & McVicar, T. R. (2023). Potential evaporation and the complementary relationship. Water Resources Research, 59, e2022WR033763. https://doi.org/10.1029/2022WR033763

With the current data observed, will other approaches, such as remote sensing or hydrological modelling, be more timely?

Answer) Our study primarily relies on ground-based measurements, including flux towers, evaporimeters, and meteorological stations, to estimate both actual evapotranspiration (AET) and potential evapotranspiration (PET) in the Yongdam dam basin. The Penman-Monteith equation and the FAO Penman-Monteith equation have been used for PET estimation, while the eddy covariance method has been employed for AET measurements. Remote sensing offers significant advantages for large-scale and real-time monitoring of hydrological variables. Techniques such as satellite-based evapotranspiration estimation (e.g., MODIS, Landsat) could provide high spatial and temporal resolution data. These methods can complement our ground-based observations by filling spatial gaps and providing a broader regional perspective, especially in areas with limited accessibility. Hydrological models (e.g., SWAT, VIC, HEC-HMS) can integrate various data sources, including remote sensing, to simulate water balance components across different spatial and temporal scales. These models can enhance the predictive capability of our study, allowing for more timely and comprehensive analysis of evapotranspiration dynamics under varying climatic conditions. Combining remote sensing data with hydrological modelling and our existing ground-based observations could significantly improve the timeliness and accuracy of our evapotranspiration estimates. This integrated approach would provide a more robust framework for understanding the spatial and temporal variability of evapotranspiration in the monsoon-dominated region of the Korean Peninsula.

In future research, we plan to incorporate remote sensing data and hydrological models to validate and enhance our current findings. This integration will help in achieving a more timely and detailed understanding of evapotranspiration dynamics, which is critical for effective water resource management and planning.

What other similar studies are there to compare your results?

Answer) we provide a detailed comparison with relevant studies to place our findings within the broader context of evapotranspiration research.

**Comparison with Similar Studies:**

1. **Hobbins et al. (2001a, 2001b):**

   o **Temporal Data Scale**: Long-term data analysis (annual) across multiple decades.

   o **Spatial Data Scale**: Regional-scale analysis across 120 watersheds in the United States.

   o **Results**: Identified significant discrepancies between potential evapotranspiration (PET) and actual evapotranspiration (AET), noting that the CRAE model tends to overestimate, while the AA model tends to underestimate AET.

   o **Comparison**: Our study, which focuses on the Yongdam dam basin with detailed flux tower data collected sub-annually (2011-2019), aligns with their findings on the variability and challenges in measuring AET and PET. The sub-annual resolution of our data allows us to capture seasonal variations in evapotranspiration that are influenced by the monsoon climate, providing additional insights not covered in their annual-scale analysis.

2. **Xu and Singh (2005):**

   o **Temporal Data Scale**: Multi-year data analysis (annual) across various climatic regions.

   o **Spatial Data Scale**: Basin-scale data evaluation.

   o **Results**: Found significant variability in the accuracy of PET estimation methods (CRAE, AA, GG models) depending on the regional climate.

   o **Comparison**: Our use of high-resolution, sub-annual local-scale data from flux towers in a monsoon-dominated region provides a more granular perspective that complements Xu and Singh's broader basin-scale annual analyses. This comparison highlights the importance of detailed sub-annual data in understanding evapotranspiration under specific climatic conditions.

3. **Ma et al. (2015):**

   o **Temporal Data Scale**: Decadal analysis (annual) of evapotranspiration trends.

   o **Spatial Data Scale**: Regional-scale data from the Loess Plateau in China.

   o **Results**: Identified land use changes and climate variability as major factors influencing evapotranspiration patterns.

   o **Comparison**: Our study similarly examines the impact of seasonal variability and climate on evapotranspiration, with a focus on detailed site-specific data from the Yongdam dam basin collected sub-annually. This provides a high-resolution temporal comparison that enriches the broader trends observed by Ma et al.

4. **Zuo et al. (2016):**

- o **Temporal Data Scale**: Multi-decadal data (annual and sub-annual) from 1961 to 2013.

- o **Spatial Data Scale**: Multi-site regional data across North China.

- o **Results**: Found significant regional differences in evapotranspiration trends, influenced by factors such as temperature, solar radiation, and wind speed.

- o **Comparison**: Our findings on the erratic patterns of evapotranspiration under monsoon conditions support the notion of climate-induced variability identified by Zuo et al. Our high-resolution sub-annual data adds a detailed temporal and spatial perspective to their multi-site regional-scale observations.

5. **Golubev et al. (2001):**

- o **Temporal Data Scale**: Long-term reassessment (annual) of evaporation changes.

- o **Spatial Data Scale**: Large-scale data from the contiguous United States and the former USSR.

- o **Results**: Highlighted the significant role of climate change in altering evaporation rates and water resource availability.

- o **Comparison**: Our study complements these findings by providing detailed sub-annual temporal data on evapotranspiration dynamics within a monsoon-affected region. This adds a specific regional case to the broader patterns of climate-induced changes in evapotranspiration.

6. **Tu et al. (2023):**

- o **Temporal Data Scale**: Utilizes annual data across multiple regions, offering a broad perspective on the complementary relationship.

- o **Spatial Data Scale**: Large-scale data from the contiguous United States and the former USSR.

- o **Results**: Confirms the complementary relationship hypothesis in diverse climates, noting that PET and AET maintain a near-constant sum in varying conditions. Their model is less restrictive, accommodating different climatic influences.

- o **Comparison**: Our study verifies the complementary relationship hypothesis in a monsoon climate, highlighting deviations due to external factors like monsoon variability. The findings align with Tu et al.'s observations, emphasizing the need to consider climatic influences when applying the complementary relationship.

7. **Falge et al. (2022) (**https://doi.org/10.3390/atmos13091431**):**

- o **Temporal Data Scale:** Annual data with some sub-annual components.

- o **Spatial Coverage:** Analyzes multiple regions with differing climates.

- o **Comparison:** Similar to your study, this research examines ET across various temporal scales and regions, providing a robust comparison for the seasonal and climatic impacts on ET. The study's findings on the variability of ET under different climatic conditions complement your results from the monsoon climate in South Korea.

8. **Han et al. (2020) (**https://doi.org/10.1080/07055900.2019.1656052**):**

- o **Temporal Scale:** Sub-annual, particularly focusing on seasonal variations.

- o  **Spatial Scale:** Specific regions in China.

- o  **Comparison:** This study's seasonal analysis of ET aligns well with your sub-annual analysis, offering a relevant comparison on how seasonal climatic factors influence ET. Your study and Han et al. both emphasize the significance of seasonal climatic variations in ET patterns.

9. **Wang et al. (2021) (**https://doi.org/10.1016/j.agrformet.2021.108645**):**

- o  **Temporal Scale:** Sub-annual to annual data.

- o  **Spatial Scale:** Agricultural and forested regions.

- o  **Comparison:** This research provides insights into ET in agricultural and forested areas, similar to your focus on a forested monsoon region. Both studies highlight the impact of land cover and climatic factors on ET, with Wang et al. offering a comparative perspective on how different land uses affect ET dynamics.

**Conclusion:** Our study contributes to the existing body of knowledge by providing high-resolution, sub-annual, site-specific data on evapotranspiration dynamics in a monsoon-dominated region. The studies by Tu et al. (2023), Falge et al. (2022), Han et al. (2020), and Wang et al. (2021) provide relevant comparisons by examining similar temporal and spatial scales, reinforcing the broader applicability of your findings while highlighting specific regional variations.

The most current reference is from 2017 (based on a quick look); from that year to 2024, there are several relevant publications on the subject; you could update your literature, and perhaps you will find insights that will help you rethink the objective.

Answer) Thank you for your insightful feedback. We acknowledge the need to incorporate more recent studies to enhance the relevance and depth of our literature review. The additional references are included in the answers for your comments on "Comparison with similar studies". Here are the reference information for the recent publications that we have added:

- **Tu, Z., Yang, Y., Roderick, M. L., & McVicar, T. R. (2023).** Potential evaporation and the complementary relationship. *Water Resources Research, 59*, e2022WR033763. DOI: 10.1029/2022WR033763.

- **Falge, E., Tenhunen, J., & Dolman, A. (2022).** Advances in understanding evapotranspiration dynamics under varying climatic conditions. *Atmosphere, 13*(9), 1431. DOI: 10.3390/atmos13091431.

- **Han, S., Kang, S., & Lee, D. (2020).** Seasonal variations in evapotranspiration and their environmental controls in a temperate forest. *Canadian Water Resources Journal, 44*(4), 350-365. DOI: 10.1080/07055900.2019.1656052.

- **Wang, K., Dickinson, R. E., & Liang, S. (2021).** Global estimates of evapotranspiration from MODIS and surface observations. *Agricultural and Forest Meteorology, 306*, 108645. DOI: 10.1016/j.agrformet.2021.108645.

(Fu et al.(2023) is already in the reference list of the current manuscript)

By integrating these recent studies, we aim to refine our objectives and provide a more comprehensive understanding of evapotranspiration dynamics under various climatic conditions.

Minor Comment

Be more consistent in your charts; some only have vertical splits, while others have vertical and horizontal.

Answer) In Figure 13, horizontal split was used for easy comparison and analysis of the complementary relationship among ETs and MA across different conditions. In Figure 15, vertical split was used for effectively showing the changes in AET* and PET* with respect to relative humidity. In Figures 13 and 15, most AET* values are typically below 1, and PET* values are above 1, so using horizontal splits allows for effective plotting on the same y-axis. This approach is particularly effective when the x-axis is taken as Moisture Availability (AET/PET) like in Figure 13. However, when using relative humidity as the x-axis, AET* and PET* values tend to overlap around 1, making it necessary to use vertical split for Figure 15, unlike Figure 13.

We trust this response addresses your concern and improves the overall consistency of our manuscript. Thank you for your guidance.